# Convergent evolution of SARS-CoV-2 XBB lineages on receptor-binding domain 455–456 synergistically enhances antibody evasion and ACE2 binding

Fanchong Jian[1,2,3☯], Leilei Feng[4,5☯], Sijie Yang[1,6☯], Yuanling Yu[2☯], Lei Wang[4,5], Weiliang Song[1,2,7], Ayijiang Yisimayi[1,2,7], Xiaosu Chen[8], Yanli Xu[9], Peng Wang[2], Lingling Yu[2], Jing Wang[2], Lu Liu[2], Xiao Niu[1,3], Jing Wang[1,2,7], Tianhe Xiao[1,10], Ran An[2], Yao Wang[2], Qingqing Gu[2], Fei Shao[2], Ronghua Jin[9], Zhongyang Shen[11], Youchun Wang[2,12], Xiangxi Wang[4,5*], Yunlong Cao[1,2*]

1 Biomedical Pioneering Innovation Center (BIOPIC), Peking University, Beijing, People's Republic of China, 2 Changping Laboratory, Beijing, People's Republic of China, 3 College of Chemistry and Molecular Engineering, Peking University, Beijing, People's Republic of China, 4 CAS Key Laboratory of Infection and Immunity, National Laboratory of Macromolecules, Institute of Biophysics, Chinese Academy of Sciences, Beijing, People's Republic of China, 5 University of Chinese Academy of Sciences, Beijing, People's Republic of China, 6 Peking-Tsinghua Center for Life Sciences, Tsinghua University, Beijing, People's Republic of China, 7 School of Life Sciences, Peking University, Beijing, People's Republic of China, 8 Institute for Immunology, College of Life Sciences, Nankai University, Tianjin, People's Republic of China, 9 Beijing Ditan Hospital, Capital Medical University, Beijing, People's Republic of China, 10 Joint Graduate Program of Peking-Tsinghua-NIBS, Academy for Advanced Interdisciplinary Studies, Peking University, Beijing, People's Republic of China, 11 Organ Transplant Center, NHC Key Laboratory for Critical Care Medicine, Tianjin First Central Hospital, Nankai University, Tianjin, People's Republic of China, 12 Institute of Medical Biology, Chinese Academy of Medical Science & Peking Union Medical College, Kunming, People's Republic of China

☯ These authors contributed equally to this work.
* xiangxi@ibp.ac.cn (XW); yunlongcao@pku.edu.cn (YC)

**Data Availability Statement:** Cryo-EM data for structures have been deposited in the Protein Data Bank (PDB) with accessions 8WRL, 8WTD, 8WRH,

## Abstract

Severe acute respiratory syndrome coronavirus 2 (SARS-CoV-2) XBB lineages have achieved dominance worldwide and keep on evolving. Convergent evolution of XBB lineages on the receptor-binding domain (RBD) L455F and F456L is observed, resulting in variants with substantial growth advantages, such as EG.5, FL.1.5.1, XBB.1.5.70, and HK.3. Here, we show that neutralizing antibody (NAb) evasion drives the convergent evolution of F456L, while the epistatic shift caused by F456L enables the subsequent convergence of L455F through ACE2 binding enhancement and further immune evasion. L455F and F456L evade RBD-targeting Class 1 public NAbs, reducing the neutralization efficacy of XBB breakthrough infection (BTI) and reinfection convalescent plasma. Importantly, L455F single substitution significantly dampens receptor binding; however, the combination of L455F and F456L forms an adjacent residue flipping, which leads to enhanced NAbs resistance and ACE2 binding affinity. The perturbed receptor-binding mode leads to the exceptional ACE2 binding and NAb evasion, as revealed by structural analyses. Our results indicate the evolution flexibility contributed by epistasis cannot be underestimated, and the evolution potential of SARS-CoV-2 RBD remains high.

8WRM, 8WRO, 8WTJ, and the Electron Microscopy Data Bank (EMDB) with accessions 37779, 37832, 37776, 37781, 37784, 37835. We used our previously published DMS data for analysis, which can be downloaded from Zenedo (DOI: 10.5281/zenodo.8373447). Other necessary data have been included in the manuscript as supplementary information.

**Funding:** Y.C. received financial support from the Ministry of Science and Technology of China (2023YFC3041500; 2023YFC3043200) (https://www.most.gov.cn/), Changping Laboratory (2021A0201; 2021D0102) (https://www.cpl.ac.cn/), and National Natural Science Foundation of China (32222030) (https://www.nsfc.gov.cn/). These sponsors or funders did not play any role in the study design, data collection and analysis, decision to publish, or preparation of the manuscript.

**Competing interests:** Y.C. is one of the inventors of the provisional patent applications for BD series antibodies, which includes SA55 (BD55-5514). Y.C. is one of the founders of Singlomics Biopharmaceuticals. Other authors declare no competing interests.

## Author summary

The continuous evolution of SARS-CoV-2 presents a challenge to global public health and the development of vaccines and treatments against COVID-19. Recently, an adjacent residue alteration, L455F+F456L, also known as "FLip", on the receptor-binding domain of the virus, which has been identified in multiple strains of the virus, alters how the virus interacts with human cells and immune response. We show that the combination of these mutations synergistically increases the virus's ability to bind to ACE2, the primary receptor on cell surfaces, enabling it to specifically escape a public type of neutralizing antibodies elicited by vaccination and infection, and the molecular mechanisms are explained by structural analyses. The enhancement of receptor binding increases the potential of the virus to further accumulate immune evasive mutations. These findings broaden our understanding of SARS-CoV-2 evolution and highlight the importance of paying attention to these ongoing antigenic drifts in the virus as we continue to develop and evaluate current antibody therapeutics and vaccines.

## Introduction

Severe acute respiratory syndrome coronavirus 2 (SARS-CoV-2) has been continuously circulating and evolving worldwide [1–4]. Since late 2022, XBB* variants, especially XBB.1.5 and other XBB derivatives with a proline on residue 486 (486P) on the receptor-binding domain (RBD) of the virus Spike glycoprotein (S) started to dominate, which demonstrated enhanced binding to human ACE2 while maintaining extremely strong capability of evading humoral immunity [5–8]. These immune-evasive lineages are still continuously accumulating more S mutations, such as R403K, V445S, L455F, F456L and K478R, that may lead to further shift in antigenicity and escape from neutralizing antibodies elicited by repeated vaccination and infection [9,10]. Some immune escape mutations, represented by F456L, even convergently appeared recently in multiple independent XBB derivative strains, such as EG.5, XBB.1.5.10, FE.1 and FD.1.1, indicating strong selection pressure due to herd immunity (Fig 1A) [11,12]. By October 2023, over 70% of newly uploaded SARS-CoV-2 sequences carry F456L mutation. Furthermore, multiple independent XBB lineages with both F456L and L455F are growing rapidly in different countries, such as XBB.1.5.70/GK.* in Brazil, the United States, and Canada, and HK.3 (EG.5.1.1.3) in China (Fig 1B and 1C) [12–14]. However, the proportion of lineages with L455F mutation but without F456L is extremely low, exhibiting no growth advantage (Fig 1C). Interestingly, L455 and F456 are two adjacent residues on the receptor-binding motif (RBM) of SARS-CoV-2 RBD, and the variant is just the "flipping" of the two residues, L455-F456 to F455-L456, also known as the "FLip" mutant (Fig 1B). These two sites are also located on a critical epitope that targeted by the public IGHV3-53/3-66 Class 1 NAbs [15–17]. Mutations on these sites are likely to escape this type of NAbs that are abundant in vaccinated and convalescent individuals, leading to substantial reduction of protection efficiency [18,19]. It is crucial to investigate the impacts on immune evasion and infection efficiency, especially for recent convalescents who recovered from XBB breakthrough infections, and the underlying mechanism of such synergistic effects that enables the unexpected advantage of L455F mutation on the basis of XBB*+F456L lineages, to explain the exceptional growth advantage of such lineages.

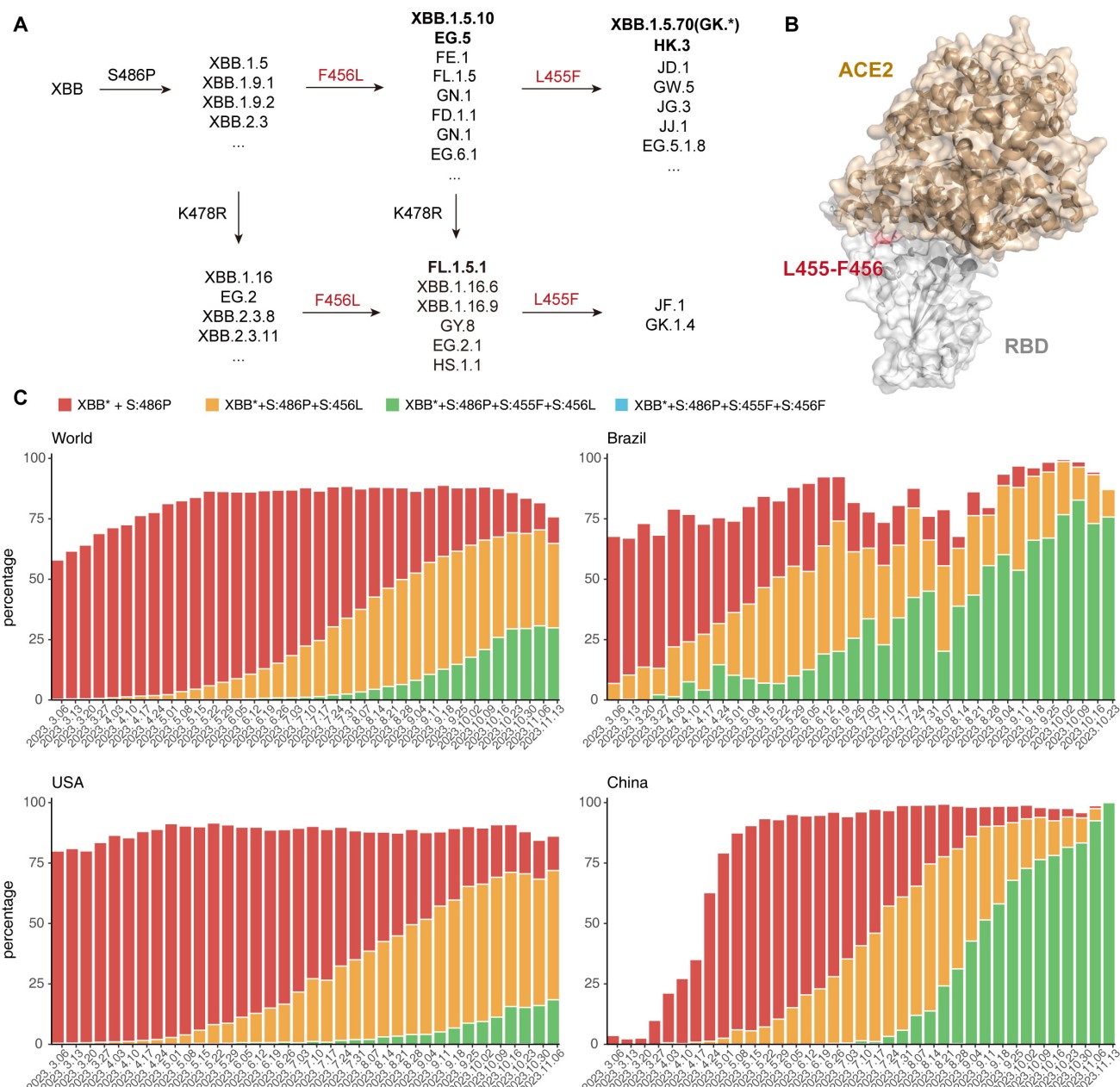

**Fig 1. Convergent evolution and circulation of SARS-CoV-2 XBB variants.** (A) RBD mutations carried by emerging XBB subvariants. L455F, F456L, and K478R are observed in multiple independent lineages, demonstrating high selection pressure and convergent evolution. Lineages are defined by Pango (https://github.com/cov-lineages/pango-designation). (B) Structural representation of the position of two convergently mutated residues L455 and F456 (colored in red) on the receptor-binding motif of XBB.1 RBD in complex with human ACE2 (colored in brown) (PDB: 8IOU). (C) Proportion of XBB subvariants since March to October 2023 with S486P (represented by XBB.1.5), S486P+F456L (represented by EG.5), or S486P+L455F+F456L (represented by XBB.1.5.70 and HK.3) among uploaded sequences in the World, Brazil, China, and United States. F456L and L455F+F456L exhibit growth advantage compared to their ancestor (XBB*+S486P). Data are collected from CoV-Spectrum (https://cov-spectrum.org).

# Results

## L455F and F456L evade convalescent plasma from XBB BTIs and reinfections

To interrogate whether L455F, F456L, and their combination leads to enhanced resistance to neutralizing antibodies elicited by vaccination and infection, we collected plasma samples

from three cohorts with distinct SARS-CoV-2 immunization histories. All participants in the three cohorts had received triple doses of CoronaVac or BBIBP/WIBP/CCIBP-CorV (inactivated vaccines based on SARS-CoV-2 ancestral strain developed by Sinovac and Sinopharm, respectively) before any known infection [20,21]. The first cohort includes participants who were breakthrough-infected by BA.5 or BF.7 (66 samples), the second infected by XBB* with S486P (mainly XBB.1.9.1/XBB.1.9.2, 27 samples), and the third experienced BA.5/BF.7 BTI before XBB*+486P reinfection (mainly XBB.1.9.1/XBB.1.9.2, 54 samples). The infected strains were inferred from the epidemiological data and uploaded SARS-CoV-2 sequences in Beijing and Tianjin, China, where the patients were recruited, during the infections (S1 Table). We evaluated their neutralization activities against SARS-CoV-2 D614G (B.1), BA.5, XBB.1.5, XBB.1.16 (XBB.1+E180V+K478R+S486P), XBB.1.5+F456L (represented by EG.5), EG.5.1 (EG.5+Q52H), XBB.1.5+F456L+K478R (represented by FL.1.5.1), and XBB.1.5+L455F+F456L (represented by XBB.1.5.70/GK.* and HK.3) spike-pseudotyped vesicular stomatitis virus (VSV). Consistent with previous studies, single BA.5/BF.7 or XBB BTI cannot elicit NAbs that efficiently neutralize all XBB subvariants due to immune imprinting, with 50% neutralization titers (NT50) lower than 100 [9,22] (Figs 2A and S1). In contrast, convalescent plasma samples from the reinfection cohort exhibit higher titers against BA.5 than D614G and neutralize XBB subvariants well, which indicates that second exposure to Omicron helps alleviate imprinting [9,18,22,23]. Notably, compared to F456L which is known to escape many NAbs elicited by Omicron reinfection, the additional L455F substitution further causes substantial evasion of plasma from the reinfection cohort [18,24,25]. Specifically, an additional L455F mutation based on XBB.1.5+F456L or EG.5 reduces plasma NT50 by 1.2-fold and 1.3-fold in the XBB BTI and reinfection cohorts, respectively, and the same degree of additional evasion could also been observed when comparing "FLip" with XBB.1.5+L455F (Fig 2A and 2B). However, the two single substitution mutants, XBB.1.5+L455F and XBB.1.5+F456L, exhibit similar level of

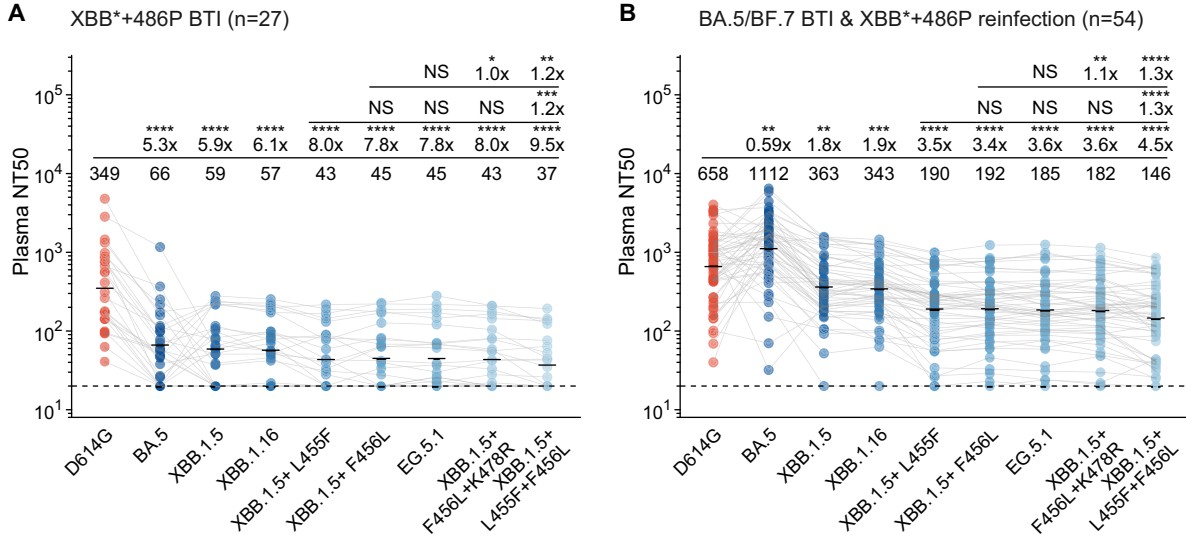

**Fig 2. Emerging XBB subvariants evade neutralization of XBB convalescent plasma.** 50% neutralization titers against SARS-CoV-2 variants of convalescent plasma from individuals who received triple doses of CoronaVac and breakthrough-infected by XBB*+486P (A, 27 samples) or BA.5/BF.7 followed by XBB*+486P reinfection (B, 54 samples). VSV-based pseudoviruses are used. Statistical significances and geometric mean titer (GMT) fold-changes are labeled in comparison with neutralization against XBB.1.5+F456L (the first line), XBB.1.5 +L455F (the second line), and D614G (the third line). Two-tailed Wilcoxon signed-rank tests of paired samples are used. *, p<0.05; **, p<0.01; ***, p<0.001; ****, p<0.0001; NS, not significant (p>0.05).

antibody evasion. Therefore, L455F further enhances the NAb evasion of XBB.1.5+F456L variants, which may contribute to the growth advantage of the "FLip" mutants. These findings indicate that L455F and F456L complement each other to achieve maximum capability of escaping NAbs, despite their both linear and spatial adjacency on the RBD.

## L455F, F456L, and their combination specifically escape the majority of Class 1 NAbs

As residues 455 and 456 on RBD are mainly recognized by Class 1 antibodies, which is also referred to as "Group A1/A2" in our previous study, we tested the pseudovirus-neutralizing activities of a panel of XBB.1.5-effective RBD-targeting monoclonal NAbs against these newly-emerged XBB subvariants, which were isolated in previous studies [9,18] (S2 Fig). The selected mAbs are expected to target L455/F456 as determined by deep mutational scanning (DMS) (Fig 3A). Mutations on L455 and F456 exhibit correlated but distinct capability of escaping these NAbs (Figs 3B and S3). Group A1 NAbs generally utilize IGHV3-53/3-66 and can cross-neutralize the early D614G (B.1) strain, while Group A2 NAbs are usually specific to Omicron lineages (S4A Fig) [18]. Consequently, L455F and F456L single substitution escape 11 and 8 out of 34 Group A1/A2 NAbs, respectively. Their combination exhibits stronger evasion, against which only 8 NAbs remain effective. Notably, 11 NAbs were not completely escaped by either L455F or F456L (IC$_{50}$ < 1 μg/mL against both mutants), but escaped by the combination, indicating the synergy of the two mutations (Figs 3C and S4B). We also evaluated the activity of neutralizing mAbs reported as drug candidates, including three that had been approved for emergency use. Two of three tested Class 1 NAbs, BD57-0129 and Omi-42, were partially evaded by L455F and L455F+F456L, while BD56-1854 remains potent [26]. The activity of Class 3 antibody S309 (Sotrovimab) is not substantially affected, remaining weak neutralization [27]. As expected, SD1-targeting antibody S3H3, and our previously reported Class 1/4 (Group F3) RBD-targeting therapeutic neutralizing antibody SA55 remain potent against all tested XBB subvariants, given that the mutated residues in the variants are not directly recognized by SA55 and S3H3 (Fig 3D) [28,29].

## Epistatic interactions of L455F and F456L on receptor binding affinity

The above results demonstrate that both the L455F and F456L mutations can confer significant resistance to neutralization by convalescent plasma, primarily mediated by escaping Class 1 NAbs. However, it remains unclear why, during natural evolution, various lineages consistently evolve F456L independently, followed by subsequent occurrence of the L455F mutation, while cases where L455F is acquired first are rarely observed. Due to the fact that residues 455–456 are also located on the RBM of RBD, we hypothesize that the effects of the L455F and F456L combination originate from their impact on the affinity to the cell surface receptor for virus entry, human ACE2 (hACE2). Previous DMS data have indicated that individual substitutions, L455F or F456L, both lead to a substantial decrease in hACE2 affinity within the BA.2 background [30]. However, as the combination of these two mutations essentially results in an adjacent residue flipping, there might be a local compensatory effect on hACE2 binding. The significant antigenic shift from BA.2 to XBB.1.5 may also alter the impacts of the two mutations on receptor binding. To validate the hypothesis, we constructed recombinant RBD subunits of XBB.1.5, XBB.1.5+L455F, XBB.1.5+F456L, and XBB.1.5+L455F+F456L ("FLip"), and determined their binding affinities to hACE2 by surface plasmon resonance (SPR) assays. The dissociation equilibrium constants (K$_D$) demonstrate 6.4 nM, 25 nM, 11 nM, and 2.0 nM for the four mutants, respectively (Fig 4A). Consistent with DMS results, L455F significantly dampens hACE2-binding affinity of XBB.1.5 RBD, and F456L also slightly weakens the

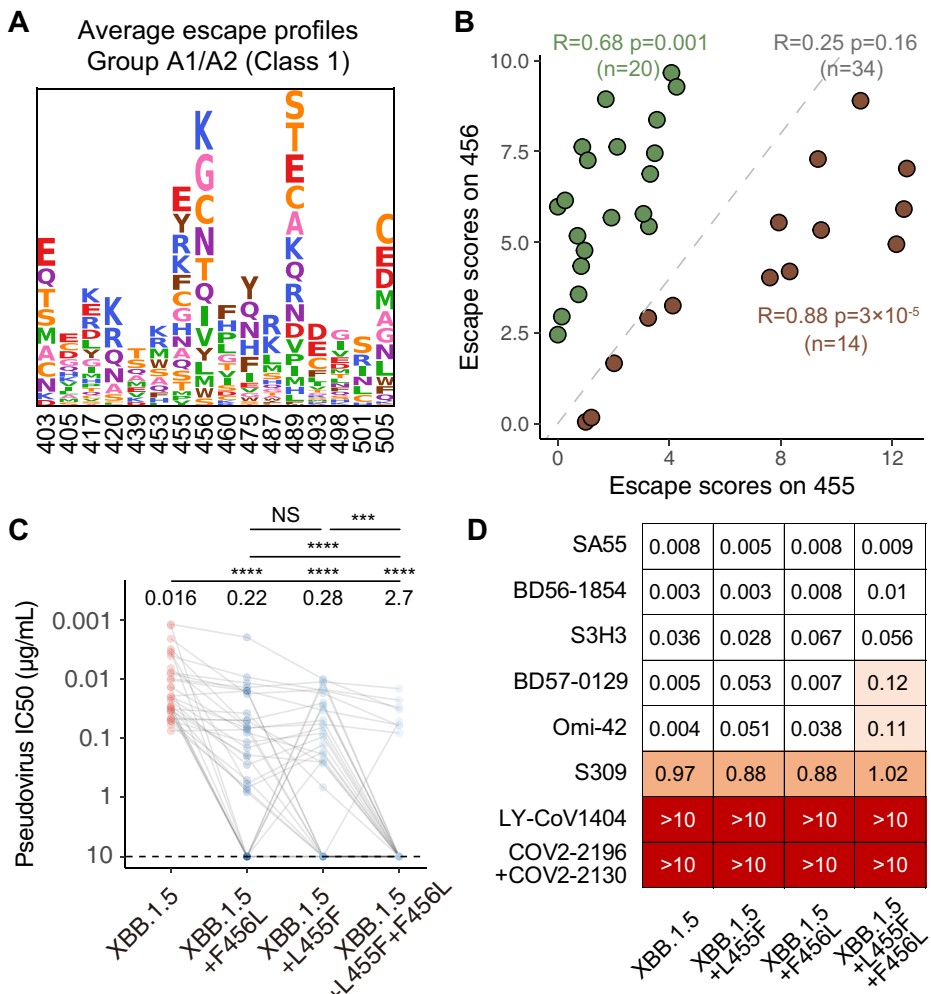

**Fig 3. L455F and F456L specifically escapes RBD Class 1 NAbs.** (A) Logo plot for average escape scores on escape hotspots of Group A1/A2 antibodies in the DMS dataset. DMS were performed based on BA.5 RBD. Letters represent amino acids. Larger heights indicate stronger escape. Amino acids are colored according to chemical properties. (B) Scatter plot shows the total escape scores on RBD site 455 and 456 of each Class 1 NAbs selected for the pseudovirus neutralization analysis. Antibodies whose escape scores on 456 are higher than that on 455 are colored green, while those with scores on 455 higher are colored brown. Pearson's correlation coefficients and corresponding p-values in green, brown, or gray indicate that the values are calculated using green, brown, or all NAbs, respectively. (C) IC50 of a panel of XBB.1.5-effective monoclonal NAbs from Group A1 and A2 (determined by DMS) against XBB.1.5, XBB.1.5 +L455F, XBB.1.5+F456L (EG.5), and XBB.1.5+L455F+F456L ("FLip") pseudovirus. Dashed lines indicate limits of detection (10 μg/mL). Geometric mean IC50 values are labeled above the points. Two-tailed Wilcoxon signed-rank tests of paired samples are used. ***, $p<0.001$; ****, $p<0.0001$; NS, not significant ($p>0.05$).(D) Neutralization of NAb drugs and selected monoclonal NAbs targeting RBD or SD1 on the Spike glycoprotein against XBB.1.5, XBB.1.5 +F456L (EG.5), and XBB.1.5+L455F+F456L ("FLip") pseudovirus. Values shown in the grids are $IC_{50}$ (μg/mL).

binding to hACE2. Surprisingly, although neither of the two mutations increases hACE2 affinity alone, their combination XBB.1.5+L455F+F456L exhibits significantly higher affinity than XBB.1.5. As for the kinetics, L455F or F456L alone does not largely affect the association kinetic constant ($k_a$), but L455F greatly fastens the dissociation ($k_d$) (S5 Fig). In contrast, "FLip" not only rescues the accelerated dissociation, but also facilitates association, synergistically improving the receptor binding at both thermodynamic and kinetic levels (Figs 4B and S5).

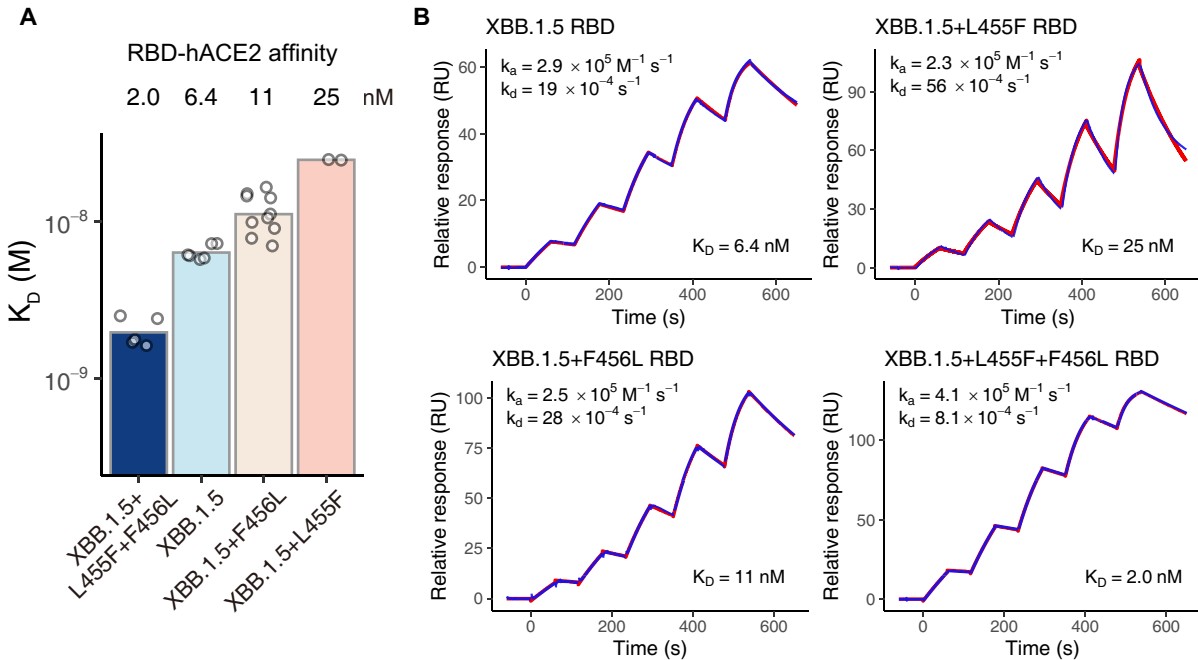

**Fig 4. "FLip" but not L455F or F456L alone enhances ACE2 binding affinity.** (A) ACE2-binding affinities of SARS-CoV-2 XBB subvariant RBD determined by SPR. Mean $K_D$ values are annotated above the bars. Each point represents a replicate. Data from all replicates are shown. SPR assays are performed in at least two independent replicates. (B) SPR sensorgrams from experiments on hACE2-binding affinity of XBB.1.5, XBB.1.5+L455F, XBB.1.5+F456L, and XBB.1.5+L455F+F456L RBD. Representative results from at least two replicates are shown. Geometric mean values of $k_a$, $k_d$, and $K_D$ over replicates are labeled on each sensorgram.

## Structural basis of the enhanced ACE2-binding affinity and antibody evasion in "FLip" variants

To elucidate the disparities in binding affinity between XBB.1.5, XBB.1.5+F456L, and XBB.1.5 +L455F+F456L ("FLip") spike proteins with ACE2, we determined the structures of these three spike proteins individually in complex with hACE2 to interrogate the conformational change on their RBD-ACE2 binding interface. As we expected, all three spike trimers mainly showed two 'RBD-open' and one 'RBD-closed' conformations with one or two ACE2 bound (Fig 5A). To further reveal the impacts of mutations of residues 455 and 456 on the interface, we determined the high-resolution structures of these three variants' RBD in complex with ACE2 at resolution of 3.3 Å, 3.0 Å, and 3.0 Å, respectively (Fig 5B). With an unambiguous electron density observed, reliable analyses of the interaction interface can be performed (S6 Fig).

Compared to XBB.1.5, F456L mutation does not substantially affect the interactions on the RBD-ACE2 interface. Although the side chain size of leucine is smaller than phenylalanine, which should slightly weaken the hydrophobic packing, the packing among RBD-L455, RBD-F456, RBD-Y489, and ACE2-F28/D30/K31 is largely kept in L456 mutant, which is in line with the slightly reduced ACE2-binding affinity of F456L (Figs 5C and S7). In the case of "FLip" (L455F+F456L), the F455 manifests a distinct side-chain orientation and conformation compared to L455 (Fig 5D). This unique pattern confers more flexible space for Q493 on RBD and the H34 on ACE2, hence enabling insertion of H34 side chain between RBD Q493 and S494, which cannot be realized in XBB.1.5 or XBB.1.5+F456L, due to potential clash between L455 and Q493 in this conformation (Fig 5E). Consequently, two additional hydrogen bonds, H34-Q493 (3.57 Å) and H34-S494 (2.57 Å), are introduced, enhancing the affinity of "FLip"

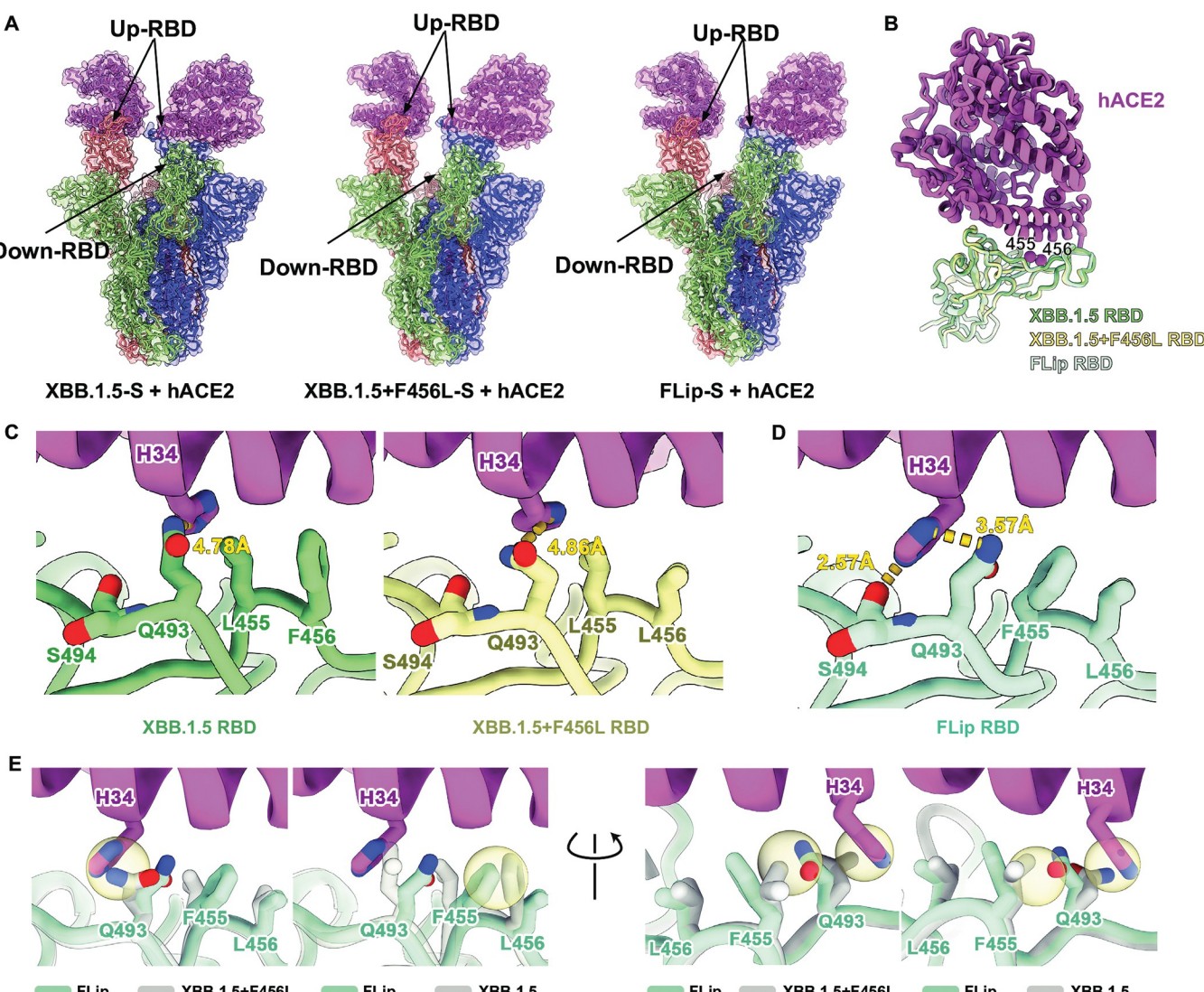

**Fig 5. Cryo-EM structure of XBB.1.5 Spike harboring L455F/F456L in complex with ACE2.** (A) Overall structure of XBB.1.5, XBB.1.5+F456L and XBB.1.5 +L455F+F456L ("FLip") prefusion-stablized spike glycoprotein in complex with human ACE2. (B) Aligned structure of XBB.1.5, XBB.1.5+F456L and XBB.1.5 +L455F+F456L RBD in complex with ACE2 after local refinement. Green, yellow and blue cartoons represent XBB.1.5, XBB.1.5+F456L and XBB.1.5+L455F +F456L RBD respectively, color pink represents ACE2. Two key mutated sites (455 and 456) are marked as magenta balls. (C) Comparison of the interfaces around site 455 and 456 of the XBB.1.5 RBD-ACE2 and XBB.1.5+F456L RBD-ACE2 complexes. (D) Interface around site 455 and 456 of the "FLip" RBD-ACE2 complex exhibits substantial conformational changes. Contacting residues are shown as sticks. Distance between atoms are shown as yellow dotted line. (E) Superposition of ACE2-binding interface structures of "FLip" with XBB.1.5 and XBB.1.5+F456L. Potential clash that limits the interface conformation in "FLip" is shown in yellow circles.

RBD to ACE2 (Fig 5D). In contrast, the distances between Q493 and H34 in XBB.1.5 and XBB.1.5+F456L are only 4.78 Å and 4.86 Å, respectively, indicating very weak interactions (Fig 5C). Further analysis concludes that F456L is a prerequisite for the fitness of "FLip". Superimposition of F456 onto "FLip" RBD could form pronounced steric clashes between F455 and F456, thereby disturbing the binding mode of "FLip" RBD-ACE2 interface (Fig 5E). Above all, the flip of L455 and F456 lead to synergistic effect of residues around and reorganize the interface between "FLip" RBD and ACE2 so that enhanced binding affinity can be obtained.

In parallel, to investigate the mechanism of L455F/F456L capability of escaping Class 1 NAbs, we conducted an analysis of the impact of the "FLip" mutation on the alteration of the

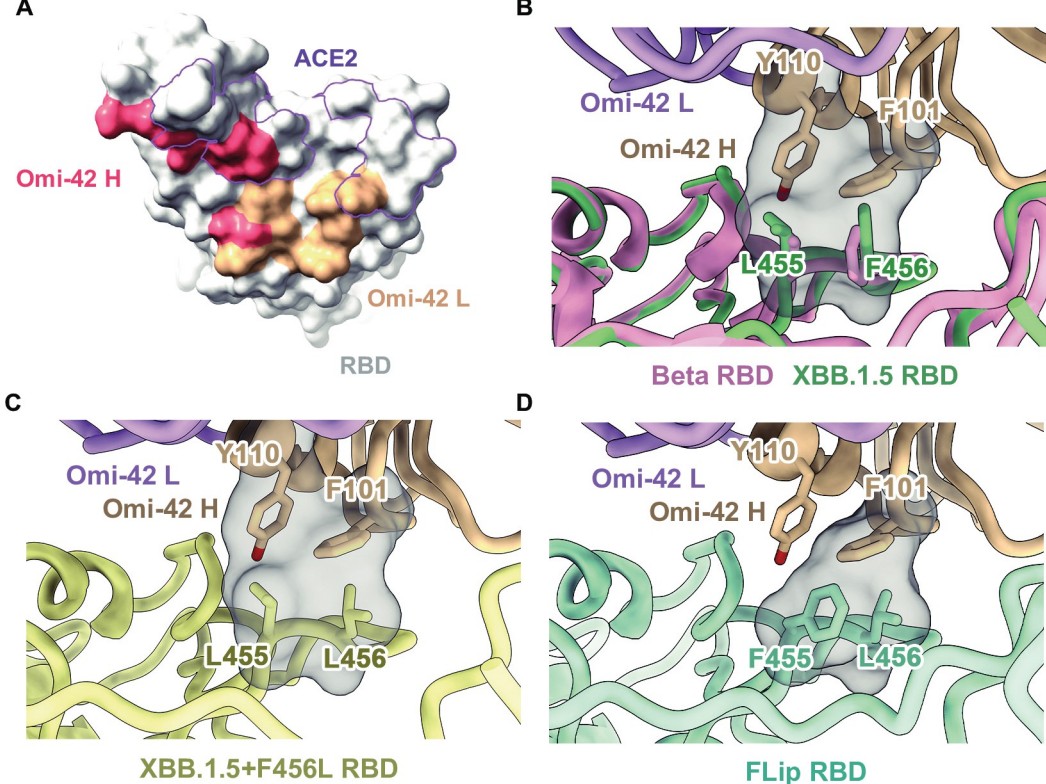

**Fig 6. Class 1 NAbs failed to mimic ACE2 binding mode.** (A) Footprint of Omi-42 heavy chain (red), light chain (orange), and ACE2 (purple contours) on the surface of RBD. (B-D) Superimposition of Omi-42 structure in complex with Beta RBD (PDB: 7ZR7) and structure of XBB.1.5 (B), XBB.1.5+F456L (C), or "FLip" (D) RBD in this study. Carbon atoms are shown as sticks, and electron density is shown in mesh. Heavy and light chain cartoons of Omi-42 are shown in grayish saffron and light purple, respectively. Colors of RBDs are the same as that in Fig 5.

neutralizing activity of NAb Omi-42, which efficiently neutralizes XBB.1.5 but not "FLip" variants [26,31,32]. Published structural models of Omi-42 in complex of Beta Spike (PDB:7ZR7) revealed that residues 455 and 456 are critically recognized by its heavy chain (Fig 6A). The superimposition of three RBDs onto the structure of mAb Omi-42 in complex with Beta RBD reveals a slight reduction in the hydrophobic patch of XBB.1.5+F456L, while Y110 of Omi-42 CDR-H3 cannot be accessible to engage in hydrophobic interaction in "FLip" RBD, resulting in a weaker interaction, consistent with the previous conclusion (Fig 6B and 6D). Notably, different Class 1 NAbs could exhibit significantly distinct interaction patterns between RBD, especially L455/F456 region, since this region is generally targeted by the highly variable CDR-H3 loop of most Class 1 NAbs, especially the public IGHV3-53/3-66 antibodies [16,17,33].

## Discussion

In this study, we evaluated the impacts of L455F and F456L, two frequently emerging adjacent mutations that convergently occur in multiple XBB sublineages, on antibody neutralization and receptor binding. We demonstrate that L455F further evades Class 1 NAbs on the XBB.1.5 +F456L basis, while single F456L or L455F substitution dampens ACE2 binding. Surprisingly, their combination, which is exactly a "FLip", i.e. "Leu-Phe" to "Phe-Leu" flipping, between adjacent residues on the ACE2 binding interface, dramatically enhances affinity to ACE2.

Together, these results explain the convergent evolution of XBB subvariants which evolve F456L and L455F in succession. Epistasis is a genetic phenomenon that the effect of one mutation is dependent on the presence of other mutations, resulting in non-additive impacts of mutations on specific functions [34,35]. Epistatic effects on the fitness of several evolving epidemic viruses, such as influenza, have been described by experiments, including DMS [36,37]. The epistatic interaction between two early SARS-CoV-2 RBD mutations on RBD, Q498R and N501Y has also been reported in a previous study [38]. Based on the ancestral SARS-CoV-2 RBD, Q498R alone slightly reduced ACE2 binding affinity, while a strong enhancement in affinity is observed for Q498R based on N501Y (RBD of Alpha VOC). The epistatic shift described here is even more striking. L455F caused strong affinity reduction on XBB.1.5, but significantly improved the hACE2-binding affinity of XBB.1.5+F456L. Both L455 and F456 are on the core of the RBD-hACE2 interface, forming a compact conformation with D30 and K31 on ACE2 [39]. Our structural analyses reveal that "FLip" mediates substantial conformation change on the RBD-hACE2 binding interface, which involves the remodeling of not only 455/456, but also Q493 on RBD and H34 on hACE2. In prior Omicron lineages, such as BA.1 and BA.2, the substitution of glutamine (Q) with arginine (R) at residue 493 on RBD significantly diminished ACE2 affinity [9,40]. A recent DMS study demonstrated that F456L is much more deleterious to ACE2 binding on R493 backbone (BA.1/BA.2) than Q493 backbone (XBB.1.5) [41]. Our structural analyses show that in the Q493 background (XBB.1.5), F456L provides a space for insertion of the side chain of K31 into the RBD, consequently introducing a hydrogen bond between K31 of ACE2 and backbone of S490 of RBD to compensate for the dampened hydrophobic packing, which cannot be achieved in the background of R493 (BA.2) due to the steric clash between ACE2-K31 and RBD-R493, making the co-occurrence of F456L and R493 deleterious (S7B Fig). It would be intriguing to investigate whether the "FLip" further accentuates the role of Q493, potentially resulting in further epistatic effects.

ACE2 mimicry has been considered to be a useful strategy for identifying and designing broad-spectrum SARS-CoV-2-neutralizing binders and antibodies [42,43]. Class 1 NAbs, such as Omi-42, usually interact with RBD L455/F456 with heavy chain CDR3 in a distinct way compared to ACE2, despite the partially overlapped footprints on RBD (Fig 6A and 6B) [26]. Similarly, most XBB.1.5-effective Class 1 NAbs failed to mimic the binding mode of ACE2 and escaped by the XBB.1.5+L455F+F456L "FLip" mutants. The emergence of L455F+F456L indicates that such receptor mimicry could be extremely difficult. To achieve the goal that mutants escaping the neutralizer also lose affinity to the receptor, the binder should not only target the same residues as those targeted by the receptor, but also mimic the binding mode of the receptor. To further investigate and explain such phenomenon, detailed analyses of the impacts these two mutations on other RBD background, particular for the variants with distinct antigenicity compared to XBB subvariants, such as BA.2.86, should be carefully evaluated in the future.

Overall, our work rationalized the emergence and circulation of XBB sublineages with F456L followed by L455F mutation, and highlighted the enhanced receptor-binding affinity and neutralizing antibody escape, which may lead to higher transmissibility and risk of breakthrough infection and reinfection. Considering the continuously increasing proportion of these variants, the efficacy of developing NAb drugs and vaccines against them should be carefully evaluated. Epistasis could substantially extend the possibility of accumulating mutations for SARS-CoV-2 RBD, leading to novel mutants with extremely high capability of escaping NAbs without great compromise on infectivity. Indeed, some additional immune-evasive mutations, including A475V, is convergently emerging on the basis of "FLip" strains. The evolutionary potential of SARS-CoV-2 RBD is still high, and should not be underestimated.

### Ethics statement

Blood samples from convalescent patients who had recovered from SARS-CoV-2 Omicron BTI or reinfection were obtained under the study protocols approved by Beijing Ditan Hospital, Capital Medical University (Ethics committee archiving No. LL-2021-024-02) and the Tianjin Municipal Health Commission, and the Ethics Committee of Tianjin First Central Hospital (Ethics committee archiving No. 2022N045KY) (S1 Table). Written informed consent, for the collection of information, storage and use of blood samples for research and data publication, was obtained from each participant.

## Materials and methods

### Patient recruitment and plasma isolation

The infections of patients in the BA.5 or BF.7 BTI cohort were confirmed between July and October 2022, during the "zero COVID" period in China. These infections were confirmed by PCR, and the viral strains of the majority of them were determined sequencing. Other samples which were not sequenced also showed strong epidemiological correlations with the sequenced samples.

Patients in the reinfection cohorts experienced first infections in December 2022 in Beijing and Tianjin. At that time, these regions were predominantly undergoing the BA.5/BF.7 wave [44]. Among the sequences from samples collected between 12/01/2022-02/01/2023, >98% of them were designated as BA.5* (excluding BQ*). Specifically, the major subtypes circulating in China at that time were BA.5.2.48* and BF.7.14*, which do not have additional mutations on RBD, and thus can be generally considered as BA.5/BF.7 (https://cov-spectrum.org/explore/China/AllSamples/from%3D2022-12-01%26to%3D2023-02-01/variants?&). Infections of patients in the XBB BTI and the second infections of patients in the reinfection cohorts were between May and June 2023, when >90% of uploaded sequences in Beijing were designated as XBB variants, and >85% were XBB* with S486P variants (https://cov-spectrum.org/explore/China/AllSamples/from%3D2023-05-01%26to%3D2023-06-15/variants?nextcladePangoLineage=XBB*&). These infections were confirmed via PCR test or antigen test.

Whole blood samples were diluted 1:1 with PBS+2% FBS and then subjected to Ficoll (Cytiva, 17-1440-03) gradient centrifugation. After centrifugation, plasma was collected from the upper layer. Plasma samples were aliquoted and stored at −20 ˚C or less and were heat-inactivated before experiments.

### Pseudovirus neutralization assay

SARS-CoV-2 variants Spike pseudovirus was prepared based on a vesicular stomatitis virus (VSV) pseudovirus packaging system. Variants' spike plasmid is constructed into pcDNA3.1 vector. G*ΔG-VSV virus (VSV G pseudotyped virus, Kerafast) and spike protein plasmid were transfected to 293T cells (American Type Culture Collection [ATCC], CRL-3216). After culture, the pseudovirus in the supernatant was harvested, filtered, aliquoted, and frozen at −80˚C for further use.

Huh-7 cell line (Japanese Collection of Research Bioresources [JCRB], 0403) was used in pseudovirus neutralization assays. Plasma samples or antibodies were serially diluted in culture media and mixed with pseudovirus, and incubated for 1 h in a 37˚C incubator with 5% $CO_2$. Digested Huh-7 cells were seeded in the antibody-virus mixture. After one day of culture in the incubator, the supernatant was discarded. D-luciferin reagent (PerkinElmer, 6066769) was added into the plates and incubated in darkness for 2 min, and cell lysis was transferred to the

detection plates. The luminescence value was detected with a microplate spectrophotometer (PerkinElmer, HH3400). IC50 was determined by a four-parameter logistic regression model.

## Antibody expression and purification

Antibody heavy and light chain genes were synthesized by GenScript, separately inserted into vector plasmids (pCMV3-CH, pCMV3-CL or pCMV3-CK) by infusion (Vazyme), and co-transfected into Expi293F cells (Thermo Fisher) using polyethylenimine transfection. The transfected cells were cultured at 36.5°C in 5% $CO_2$ and 175 rpm. for 6–10 days. The expression fluid was collected and centrifuged. After centrifugation, supernatants containing monoclonal antibodies were purified using Protein A magnetic beads (Genscript), and the purified samples were verified by SDS-PAGE.

## Recombinant RBD expression and purification

DNA fragments that encode SARS-CoV-2 variant RBD (Spike 319–541) were codon-optimized for human cell expression and synthesized by Genscript. His-AVI tags were added at the end of the fragments. The fragments were then inserted into pCMV3 vector through infusion (Vazyme). The recombination products were transformed into E. coli DH5α competent cells (Tsingke). Colonies with the desired plasmids were confirmed by Sanger sequencing (Azenta) and cultured for plasmid extraction (CWBIO). 293F cells were transfected with the constructed plasmids and cultured for 6 days. Products were purified using Ni-NTA columns (Changzhou Smart-lifesciences, SA005100) and the purified samples were verified by SDS-PAGE.

## Surface plasmon resonance

SPR experiments were performed on the Biacore 8K (Cytiva). Human ACE2 with Fc tag was immobilized onto Protein A sensor chips (Cytiva). Purified SARS-CoV-2 variant RBDs were prepared in serial dilutions (6.25, 12.5, 25, 50, and 100nM) and injected over the sensor chips. The response units were recorded by Biacore 8K Evaluation Software 3.0 (Cytiva) at room temperature, and the raw data curves were fitted to a 1:1 binding model using Biacore 8K Evaluation Software 3.0 (Cytiva).

## Protein expression and purification for Cryo-EM

The plasmid encoding the full-length spike (S) protein (residues 1–1208, GenBank: MN908947) was used as the template for the construction of the S gene of of XBB.1.5 (T19I, Δ24–26, A27S, V83A, G142D,Δ144, H146Q, Q183E, V213E, G252V, G339H, R346T, L368I, S371F, S373P, S375F, T376A, D405N, R408S, K417N, N440K, V445P, G446S, N460K, S477N, T478K, E484A, F486P, F490S, R493Q, Q498R, N501Y, Y505H, D614G, H655Y, N679K, P681H, N764K, D796Y, Q954H, N969K) XBB.1.5.10 (XBB.1.5+F456L) (T19I,Δ24–26, A27S, V83A, G142D,Δ144, H146Q, Q183E, V213E, G252V, G339H, R346T, L368I, S371F, S373P, S375F, T376A, D405N, R408S, K417N, N440K, V445P, G446S, L455F, N460K, S477N, T478K, E484A, F486P, F490S, R493Q, Q498R, N501Y, Y505H, D614G, H655Y, N679K, P681H, N764K, D796Y, Q954H, N969K) and XBB.1.5.70 (XBB.1.5+L455F+F456L) (T19I,Δ24–26, A27S, V83A, G142D,Δ144, H146Q, Q183E, V213E, G252V, G339H, R346T, L368I, S371F, S373P, S375F, T376A, D405N, R408S, K417N, N440K, V445P, G446S, L455F, F456L, N460K, S477N, T478K, E484A, F486P, F490S, R493Q, Q498R, N501Y, Y505H, D614G, H655Y, N679K, P681H, N764K, D796Y, Q954H, N969K) by overlapping PCR. All the RBDs are constructed from the S gene plasmids. The S gene was constructed into the vector pCAGGS with a T4 fibritin trimerization motif, a HRV3C protease site and a Twin-Strep-tag at the C terminus

and was mutated as previously described [45]. To obtain the protein, the expression vector was transiently transfected into HEK293F cells grown in suspension at 37°C in a rotating, humidified incubator supplied with 8% $CO_2$ and maintained at 130 rpm. After incubation for 72h, the supernatant was harvested, concentrated, and exchanged into the binding buffer by tangential flow filtration cassette. The S proteins were then separated by chromatography using resin attached with streptavidin and further purified by size exclusion chromatography using a Superose 6 10/300(GE Healthcare) in 20 mM Tris, 200mM NaCl, pH8.0.

## Cryo-EM sample collection, data acquisition and structure determination

The cryo-EM samples of S trimers in complex with ACE2 with a molar ratio of 1:4 (S protein to ACE2) on ice to obtain S-ACE2-complex. Then, the complex was deposited onto the freshly glow-discharged grids (C-flat 1.2/1.3 Au). After 6 seconds' blotting in 100% relative humidity, the grid was plunged into liquid ethane automatically by Vitrobot (FEI). Movies (32 frames, each 0.2 s, total dose of 60 e-Å-2) were recorded using a K3 Summit direct detector with a defocus range between 1.5–2.7 μm. Automated single particle data acquisition was carried out by SerialEM, with a calibrated magnification of 75,000, yielding a final pixel size of 1.07 Å. A total of 7721, 3627 and 3040 micrographs for XBB.1.5, XBB.1.5.10 (XBB.1.5+F456L) and XBB.1.5.70 (XBB.1.5+L455F+F456L) S trimers mixed with ACE2 were collected. cryoSPARC was used to correct beam induced motion and average frames. Then, the defocus value of each micrograph was estimated by patch CTF estimation. 2231681, 607934, 981307 particles of XBB.1.5, XBB.1.5.10 and XBB.1.5.70 S-ACE2 complexes were autopicked and extracted for further 2D classification and hetero-refinement. After that, 541205, 206061, 175934 particles of XBB.1.5, XBB.1.5.10 and XBB.1.5.70 S-ACE2 complexes were used for homo-refinement in cryoSPARC for the final cryo-EM density.

To improve the resolution of the binding surface of RBD-ACE2 resolution, the cryo-EM samples of three RBDs in complex with ACE2 were also deposited in the same way, with an Fab SN1600 added. The cryo-EM samples of RBDs in complex with ACE2 and SN1600 were mixed in a molar ratio of 1:1.2:1.2 (RDB: ACE2: SN1600). Movies (32 frames, each 0.2 s, total dose of 60 e-Å-2) were recorded using a Falcon 4 Summit direct detector with a defocus range between 1.5–2.7 μm. Automated single particle data acquisition was carried out by EPU, with a calibrated magnification of 96,000, yielding a final pixel size of 0.808 Å. A total of 7135, 7106 and 8159 micrographs for XBB.1.5, XBB.1.5.10 and XBB.1.5.70 RBD mixed with ACE2 and SN1600 complexes were collected. cryoSPARC was used to correct beam induced motion and average frames. Then, the defocus value of each micrograph was estimated by patch CTF estimation. 989244, 1542948, 4207904 particles of XBB.1.5, XBB.1.5.10 and XBB.1.5.70 RBD-ACE2-SN1600 complexes were autopicked and extracted for further 2D classification and hetero-refinement. After that, 989244, 441463, 345756 particles of XBB.1.5, XBB.1.5.10 and XBB.1.5.70 RBD-ACE2-SN1600 complexes were used for homo-refinement in cryoSPARC for the final cryo-EM density.

The resolutions were evaluated on the basis of the gold-standard Fourier shell correction (threshold = 0.143) and evaluated by ResMap. All dataset processing workflows are shown in S6 Fig.

## Structural model fitting and refinement

The atom models of the complex were first fitting the chain of the apo (PDB: 7XNQ) and Fab (heavy chain: 7E5Y, light chain: 7RU3) into the obtained cryo-EM density by Chimera. Then the structure was manually adjusted and corrected according to the protein sequences and density in Coot, real-space refinement was performed by Phenix.

## Supporting information

**S1 Table. Information of SARS-CoV-2 patients involved in the study.**
(XLSX)

**S1 Fig. BA.5 or BF.7 BTI does not elicit strong neutralization against new variants.** 50% neutralization titers against SARS-CoV-2 variants of convalescent plasma from individuals who received triple doses of CoronaVac and breakthrough-infected by BA.5 or BF.7 (66 samples). VSV-based pseudoviruses are used. Statistical significances and geometric mean titer (GMT) fold-changes are labeled in comparison with neutralization against XBB.1.5+F456L (the first line) and D614G (the second line). Two-tailed Wilcoxon signed-rank tests of paired samples are used. *, p<0.05; **, p<0.01; ***, p<0.001; ****, p<0.0001; NS, not significant (p>0.05).
(PDF)

**S2 Fig. Information of XBB.1.5-neutralizing Class 1 mAbs involved in the study.** The antibody names, source cohorts, VDJ genes utilization, somatic hypermutation (SHM) ratio, and CDR lengths are shown in the table. Antibodies with the public IGHV3-53/3-66 heavy chain V genes are marked in blue background.
(PDF)

**S3 Fig. L455F and F456L show correlated but distinct evasion patterns against mAbs in the DMS dataset.** Paired correlation plots show the pairwise relationship between the escape score of L455F/F456L and total escape scores on 455/456. Antibodies are split into two groups according to their total escape scores on 455/456 as described in Fig 3B.
(PDF)

**S4 Fig. Neutralization activities of Class 1 NAbs against variant pseudoviruses.** (A) IC50 (μg/mL) against D614G, XBB.1.5, XBB.1.5+F456L, XBB.1.5+L455F, and "FLip" (XBB.1.5 +L455F+F456L) pseudoviruses using selected XBB.1.5-effective Class 1 monoclonal NAbs. (B) Fold changes of IC50 values compared to IC50 against XBB.1.5 pseudovirus for the Class 1 NAbs. "/" indicates complete escape by the mutant.
(PDF)

**S5 Fig. Kinetic parameters for RBD-ACE2 binding from SPR assays.** Association (A) and dissociation (B) constants of the binding kinetics between XBB variants RBD and hACE2 determined by SPR. Each dot indicates one independent SPR measurement. Geometric mean values are shown as bars.
(PDF)

**S6 Fig. Workflow for cryo-EM structural models.** Workflow to generate refined structural model of XBB.1.5 Spike, XBB.1.5+F456L (XBB.1.5.10) Spike, XBB.1.5+L455F+F456L (XBB.1.5.70) Spike, XBB.1.5 RBD, XBB.1.5.10 RBD and XBB.1.5.70 RBD in complex of ACE2.
(PDF)

**S7 Fig. F456L remodels the interaction between RBD and ACE2.** (A) The electron density of XBB.1.5 and XBB.1.5+F456L RBD-ACE2 complex around site 455 and 456 in detail, carbon atoms are shown as sticks, density is shown in mesh, and the color scheme is the same as that in Fig 5C. (B) Superimposition of BA.2 RBD-ACE2 (PDB:7ZF7) and XBB.1.5+F456L RBD-ACE2 complex structure. Hydrogen bonds are shown as yellow dashed lines. Potential steric clash is shown in yellow circles.
(PDF)

## Acknowledgments

We are grateful to scientists in the community for their continuous tracking of SARS-CoV-2 variants and helpful discussion, including Jesse Bloom, Daniele Focosi, Ryan Hisner, Cornelius Roemer, Federico Gueli, Tom Peacock, Raj Rajnarayanan, and many other researchers. We thank all volunteers for providing the blood samples.

## Author Contributions

**Conceptualization:** Yunlong Cao.

**Data curation:** Fanchong Jian, Leilei Feng, Sijie Yang.

**Formal analysis:** Fanchong Jian, Leilei Feng, Weiliang Song.

**Funding acquisition:** Yunlong Cao.

**Investigation:** Fanchong Jian, Leilei Feng, Yuanling Yu, Lei Wang, Ayijiang Yisimayi, Peng Wang, Lingling Yu, Jing Wang, Lu Liu, Xiao Niu, Jing Wang, Tianhe Xiao, Ran An, Yao Wang.

**Methodology:** Yuanling Yu, Yunlong Cao.

**Project administration:** Fei Shao.

**Resources:** Yuanling Yu, Xiaosu Chen, Yanli Xu, Ronghua Jin, Zhongyang Shen, Youchun Wang.

**Supervision:** Xiangxi Wang, Yunlong Cao.

**Visualization:** Fanchong Jian, Leilei Feng, Sijie Yang, Weiliang Song.

**Writing – original draft:** Fanchong Jian.

**Writing – review & editing:** Fanchong Jian, Qingqing Gu, Yunlong Cao.

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
