## [Decision Letter · Decision Letter 0]

19 Nov 2023

Dear Dr. cao,

Thank you very much for submitting your manuscript "Convergent evolution of SARS-CoV-2 XBB lineages on receptor-binding domain 455-456 synergistically enhances antibody evasion and ACE2 binding" for consideration at PLOS Pathogens. As with all papers reviewed by the journal, your manuscript was reviewed by members of the editorial board and by several independent reviewers. In light of the reviews (below this email), we would like to invite the resubmission of a significantly-revised version that takes into account the reviewers' comments.

We cannot make any decision about publication until we have seen the revised manuscript and your response to the reviewers' comments. Your revised manuscript is also likely to be sent to reviewers for further evaluation.

Sincerely,

Xiu-Feng Wan

Guest Editor

PLOS Pathogens

Guangxiang Luo

Section Editor

PLOS Pathogens

Kasturi Haldar

Editor-in-Chief

PLOS Pathogens

orcid.org/0000-0001-5065-158X

Michael Malim

Editor-in-Chief

PLOS Pathogens

orcid.org/0000-0002-7699-2064

Reviewer's Responses to Questions

**Part I - Summary**

Reviewer #1: L455F mutation has been acquired by multiple independent XBB derivative strains that contain F456L mutation. This study aims to investigate the interaction between L455F and F456L. The authors show that L455F+F456L can better escape both polyclonal plasma samples and monoclonal antibodies. They also show that epistasis exists between L455F and F456L in terms of hACE2 binding. Mechanism of this epistasis is revealed by cryo-EM analysis. Overall, this is a nice case study on a pair of RBD mutation and demonstrates the role of epistasis in the ongoing SARS-CoV-2 evolution.

Reviewer #2: This a well done and timely study explaining the sign epistasis in ACE2 affinity that appears to have promoted the emergence of the L455F / F456L double mutant. The paper nicely shows that these are both antibody escape

mutations (partially but not completely redundant in the antibodies they escape), and the reason they probably

have emerged so frequently as double mutants (F456L followed by L455F) is that while each mutation individual

ly impairs ACE2 affinity, together they enhance it.

Given the recent evolution of SARS-CoV-2 these are important results. In fact, many of the findings have admir

ably already been shared by the authors on Twitter to great interest to the relevant scientific community, and

this paper nicely fully reports the results in a comprehensive study.

I strongly support publication of this paper, and just have the two minor comments below:

The authors could discuss a bit more the role of Q493 (versus R493). Other work (Taylor and Starr, DOI 10.1101/2023.09.11.557279, figure 3D; Dadonaite et al, DOI 10.1101/2023.11.13.566961, figure 1C) have shown that F456

L is much more deleterious on the R493 background of BA.2 than Q493 backgrounds (like XBB.1.5). It would appea

r that the structural biology work presented here offers a partial explanation for that observation, and this

could be briefly mentioned.

Figure 3c, the word "effective" is mis-spelled in the legend.

**Part II – Major Issues: Key Experiments Required for Acceptance**

Reviewer #1: 1. Lines 60-61: “However, the lineages with L455F mutation but without F456L did not exhibit any growth advantage [13].” Ref #13 is a bioinformatics paper and does not mention L455F mutation. This statement needs to be supported by additional references or data.

2. Line 97: “L455F and F456L complement each other”. This statement is not true if XBB1.5 + L455F achieves the same or even larger degree of escape compared to XBB1.5 + L455F + F456L. However, XBB1.5 + L455F was not included in the neutralization assay in Figure 2. So there is insufficient evidence to claim that F456L complements L455F.

3. Lines 160-162: “This unique pattern confers more flexible space for Q493 on RBD and the H34 on ACE2, hence enabling insertion of H34 side chain between RBD Q493 and S494, which cannot be realized in XBB.1.5 or XBB.1.5+F456L.” There is no data to support the claim that Q493 cannot adopt the rotamer seen in Figure 5D in XBB.1.5 or XBB.1.5+F456L. This concern can potentially be addressed by performing an analysis similar to Figure 5E.

Reviewer #2: (No Response)

**Part III – Minor Issues: Editorial and Data Presentation Modifications**

Reviewer #1: 1. Line 110: “Notably, 11 NAbs were not completely escaped by either L455F or F456L …” The authors may want to describe the definition of “completely escaped”. My understanding is that it means no detected neutralization activity at 10 μg/mL.

2. Line 150: “both three spike trimers” should be “all three spike trimers”.

3. Line 155: “F456L mutation does not substantially affect the interactions on the RBD-ACE2 interface, keeping the hydrophobic packing … ” However, F456L should weaken the hydrophobic packing with hACE2 because L is shorter and less bulkier than F.

4. Line 158: “… negligible impact on ACE2-binding affinity of F456L (Figure 5C and S7).” According to Figure 4A, F456L weakened the binding affinity by almost 2-fold, which is a small but not negligible impact. In line 138, the author also mentioned that “F456L also slightly weakens the binding to hACE2”. So the phrase “negligible impact” seem inappropriate. This concern is related to the previous comment.

5. Line 159: “distinct rotamer pattern”. Please clarify what this term means.

6. Lines 217-219: “detailed experimental structural analyses and computational simulation of XBB.1.5 RBD with these two mutations in complex with ACE2 should be necessary in the future.” I am a bit confused here because the structure of XBB.1.5 RBD in complex with ACE2 has already been determined in this study.

7. Line 255, the meaning of “evolution flexibility” is unclear.

Reviewer #2: (No Response)

PLOS authors have the option to publish the peer review history of their article (what does this mean?). If published, this will include your full peer review and any attached files.

Reviewer #1: No

Reviewer #2: **Yes: **Jesse Bloom
---

## [Decision Letter · Decision Letter 1]

28 Nov 2023

Dear Dr. Cao,

We are pleased to inform you that your manuscript 'Convergent evolution of SARS-CoV-2 XBB lineages on receptor-binding domain 455-456 synergistically enhances antibody evasion and ACE2 binding' has been provisionally accepted for publication in PLOS Pathogens.

Best regards,

Xiu-Feng Wan

Guest Editor

PLOS Pathogens

Guangxiang Luo

Section Editor

PLOS Pathogens

Kasturi Haldar

Editor-in-Chief

PLOS Pathogens

orcid.org/0000-0001-5065-158X

Michael Malim

Editor-in-Chief

PLOS Pathogens

orcid.org/0000-0002-7699-2064

Reviewer Comments (if any, and for reference):

Reviewer's Responses to Questions

**Part I - Summary**

Reviewer #1: The authors have addressed all my previous concerns.

Reviewer #2: The authors have satisfactorily addressed the questions by me and the other reviewer.

**Part II – Major Issues: Key Experiments Required for Acceptance**

Reviewer #1: (No Response)

Reviewer #2: (No Response)

**Part III – Minor Issues: Editorial and Data Presentation Modifications**

Reviewer #1: (No Response)

Reviewer #2: (No Response)

PLOS authors have the option to publish the peer review history of their article (what does this mean?). If published, this will include your full peer review and any attached files.

Reviewer #1: No

Reviewer #2: No

---

## [Editor Report · Acceptance letter]

13 Dec 2023

Dear Associate Investigator Cao,

We are delighted to inform you that your manuscript, "Convergent evolution of SARS-CoV-2 XBB lineages on receptor-binding domain 455-456 synergistically enhances antibody evasion and ACE2 binding," has been formally accepted for publication in PLOS Pathogens.

Best regards,

Michael Malim

Editor-in-Chief

PLOS Pathogens

orcid.org/0000-0002-7699-2064